# A Fast and Robust Lane Detection Method Based on Semantic Segmentation and Optical Flow Estimation

**DOI:** 10.3390/s21020400

**Published:** 2021-01-08

**Authors:** Sheng Lu, Zhaojie Luo, Feng Gao, Mingjie Liu, KyungHi Chang, Changhao Piao

**Affiliations:** 1School of Advanced Manufacturing Engineering, Chongqing University of Posts and Telecommunications, Chongqing 400065, China; lusheng@cqupt.edu.cn (S.L.); S182101022@stu.cqupt.edu.cn (Z.L.); 2School of Automation, Chongqing University of Posts and Telecommunications, Chongqing 400065, China; S170301013@stu.cqupt.edu.cn (F.G.); liumj@cqupt.edu.cn (M.L.); 3Department of Electrical and Computer Engineering, Inha University, 100 Inha-ro Michuhol-gu, Incheon 22212, Korea; khchang@inha.ac.kr

**Keywords:** automated driving, lane detection, semantic segmentation, optical flow estimation, coordinate mapping

## Abstract

Lane detection is a significant technology for autonomous driving. In recent years, a number of lane detection methods have been proposed. However, the performance of fast and slim methods is not satisfactory in sophisticated scenarios and some robust methods are not fast enough. Consequently, we proposed a fast and robust lane detection method by combining a semantic segmentation network and an optical flow estimation network. Specifically, the whole research was divided into three parts: lane segmentation, lane discrimination, and mapping. In terms of lane segmentation, a robust semantic segmentation network was proposed to segment key frames and a fast and slim optical flow estimation network was used to track non-key frames. In the second part, density-based spatial clustering of applications with noise (DBSCAN) was adopted to discriminate lanes. Ultimately, we proposed a mapping method to map lane pixels from pixel coordinate system to camera coordinate system and fit lane curves in the camera coordinate system that are able to provide feedback for autonomous driving. Experimental results verified that the proposed method can speed up robust semantic segmentation network by three times at most and the accuracy fell 2% at most. In the best of circumstances, the result of the lane curve verified that the feedback error was 3%.

## 1. Introduction

Lane detection, also named as lane line detection in this paper, is one of the key technologies in autonomous driving, which is the precondition of feedback for control and path planning. Researchers take a combination of handcrafted features, highly specialized and heuristic, to detect lanes before deep learning-based methods appear. These methods can be divided into two categories, geometric modeling and energy minimization.

Most methods in geometric modeling follow a two-step solution including edge detection and curve fitting. Different kinds of gradient filters are used to solve edge detection. In [1], images processed by inverse perspective transformation were calculated by Gaussian filter. In [2,3], Gabor filter and Steerable filter were used to detect edge of lane lines. In addition, the structure tensor [4], base color features [5], ridge features [6], and the bar filter [7] were also used to detect lane line edges. For curve fitting, Hough Transform (HT) and extended HT method such as polar randomized HT [8] are usually utilized. For energy minimization-based method, conditional random field (CRF) is commonly investigated. In [9], CRF was proposed to detect multiple lanes by establishing an optimal association of lane marks in sophisticated scenarios. 

With the development of large-scale visual data sets and increased computation power, deep neural network (DNN), particularly convolutional neural network (CNN), has demonstrated record breaking performance in lane detection. Neven [10] combined efficient neural network (ENet) [11] with discriminative loss function [12] and cluster was used to detect lane line. It is fast and creative; however, the method is not robust enough for a complex scene, since ENet is a fast and slim semantic segmentation network. In [13], the embedding loss driven generative adversarial networks (EL-GAN) was proposed, which detects lanes accurately and can avoid marking boundaries. However, compared with a high-precision model, it is still not good enough.

As shown in Figure 1, continuous frames are similar in lane detection. It is not wise to detect every frame using the same method. Considering this, our ideal is that difficult frames are detected by a robust model and simple frames are detected by a fast model. The current frame has a big difference with the previous one as difficult frame. Otherwise, it is a simple frame, which can be distinguished by the adaptive scheduling threshold that is a floating number. Therefore, to improve detection speed and retain the robustness, we proposed a method that sufficiently utilizes the spatio-temporal information from frame to frame in lane segmentation. The whole model consisted of three sub-networks. The semantic segmentation network was responsible for detecting lane robustly, which is just applied to difficult frames. The optical flow estimation network was to find out the spatio-temporal information and track lanes fast. The adaptive scheduling network was to schedule the optical flow estimation network and the segmentation network adaptively. Due to optical flow estimation network and adaptive scheduling network, only a few frames were selected as key frames, which are segmented by semantic segmentation network, and others were tracked by optical flow estimation network, which can highly improve the processing time. In addition, we utilized DBSCAN and proposed a mapping method to get the lane curves in the camera coordinate system, which is able to provide feedback for automatic driving.

## 2. Materials and Methods

The main content for lane detection was divided into three parts including preprocessing, lane detection, and post-processing, as shown in Figure 2. In Section 2, the content includes preprocessing and lane detection. In Section 2.1, the frame is clipped based on the region of interest. This operation can reduce computation as far as possible. In Section 2.2, a lane detection method based on semantic segmentation and optical flow estimation is proposed. In Section 2.3, lanes are tracked based on bilinear interpolation. In Section 2.4, we designed an optimizing convolutional layer to optimize tracking results. In Section 2.5, an extremely small neural network was designed to schedule the two networks. In Section 2.6, an algorithm to map the abscissa of pixels from the pixel coordinate system to camera coordinate system is proposed.

### 2.1. Preprocessing

To ignore the influence of background and improve the efficiency of computer hardware, the image was clipped in a region of interest to lane detection. In this paper, the input image size of the experiment was 640 × 360, and the size of the region of interest was 640 × 160. According to the changing of image size, the computation of lane detection was only 44.4% compared with the original image. As shown in Figure 3, the area of interest was below the red line.

### 2.2. Lane Discrimination

As shown in Figure 4, the whole model can be divided into a segmentation network, an optical flow tracking module, and adaptive scheduling network. In our study, a robust semantic segmentation network was used to product a robust binary lane segmentation result at the pixel level. The optical flow tracking module, including an optical flow estimation network, a wrapping unit based on bilinear interpolationm and an optimizing convolution layer, was used to find out spatio-temporal information so that a lane segmentation result could be rapidly produced. We designed a tiny neural network that was named as adaptive scheduling network (ASN), which was used to decide working module (segmentation network or optical flow tracking module) based on the optical flow estimation network’s down-sampling feature map. The pipeline of lane segmentation module and some definitions are shown in Figure 4.

**Definition** **1.**
*Adaptive scheduling threshold. The adaptive scheduling threshold t is a floating number that is used to schedule the segmentation network and the optical flow estimation tracking module.*


**Definition** **2.**
*Adaptive scheduling network output value. The adaptive scheduling network output, a value of pr, is the output of ASN. Compared with t, if pr is greater than t, optical flow tracking module will work. Otherwise, segmentation network does.*


**Definition** **3.**
*Key frame. The key frame is generated in two cases. The first frame should be the key frame. In addition, if the current frame c is scheduled to be segmented by segmentation network, it will replace the former key frame as the new key frame, donated as k.*


When the first frame is input into the lane segmentation model, it will be used as the key frame *k* for lane segmentation. Lane segmentation on it is accomplished by segmentation network. Then, the input frame will be scheduled by ASN. The current frame *c* and key frame *k* will be input to the optical flow estimation network together. Optical flow estimation network is a typical structure of encoder and decoder. The encoder is responsible for down-sampling and produces a feature map. The feature map will be used as input to the ASN, getting *pr*. If *pr* is greater than *t*, it means that it is similar to the key frame. Lane segmentation can be accomplished rapidly by optical flow tracking module. On the contrary, the current frame *c* is taken as the new key frame *k* and input to segmentation network for lane segmentation.

#### 2.2.1. Segmentation Network

Deeplabv3plus [14] was adopted as the segmentation network in our lane segmentation model. In fact, any semantic segmentation network can be used in this place. As long as the segmentation network is faster than the optical flow tracking module, the lane segmentation speed will be improved in different degrees. The reason why Deeplabv3plus was adopted here is that it is the most robust semantic segmentation network at present. Although Deeplabv3plus is slow in single-frame segmentation and is difficult to use in autonomous vehicles directly, it achieved higher speed and retained robustness in our model.

The result of lane segmentation cannot be utilized for curve fitting, since it is hard to confirm which lane those pixels belong to. Therefore, DBSCAN was adopted to discriminate the lanes. The discrimination effect is shown in the Figure 5.

#### 2.2.2. Optical Flow Estimation Network and Training Method

In order to speed lane segmentation up and retain robustness, we combined an optical flow estimation network with a wrapping unit based on bilinear interpolation and an optimizing convolutional layer. FlowNet [15] was adopted as optical flow estimation network. The structure of FlowNet is simple and its running speed is fast. It has two encoder network structures: FlowNetSimple (FlowNetS) and FlowNetCorrelation (FlowNetC). We adopted FlowNetS here and reduced the number of channels to one-third as the optical flow estimation network. As with the segmentation network, any optical flow estimation network can be used here, theoretically. 

The training of the deep learning model needs a lot of marked data. However, there are only a few data sets available for the training of optical flow estimation network, and none of the data sets relates to lane. Therefore, it is not feasible to train an optical flow estimation network according to conventional methods. The DDFlow [16], an unsupervised optical flow estimation network training method proposed by Liu P. [16], was adopted to carry out unsupervised training on the optical flow estimation network in this paper. The unsupervised training method does not need any marked data and only needs the original data to complete the training. Please refer to the literature [16] for the specific method.

### 2.3. Wrapping Unit Based on Bilinear Interpolation

After optical flow estimation result is obtained, the feature of the original key frame can be directly moved to the corresponding position of the current frame by wrapping unit based on bilinear interpolation. Compared with single-frame detection by using the segmentation network, this operation is more flexible. 

Figure 6 is the schematic diagram of the algorithm where the horizontal axis is denoted as *u* and the vertical axis as *v*. After inputting key frame *k* and current one *c* to the optical flow estimation network, it will output an optical flow estimation matrix whose channel is two, which is the same size as input. The element of the first channel of the optical flow estimation matrix is denoted as *x*_(*u*, *v*)_, which means the displacement of the pixel of the current frame (*u*, *v*) in the *u*-axis is relative to the key frame. Similarly, *y*_(*u*, *v*)_ represents the displacement of the pixel of the current frame (*u*, *v*) relative to the key frame in the *v*-axis. Assuming that the pixel coordinate to be tracked is (*u*_0_, *v*_0_) in the current frame and its corresponding displacements relative to the key frame on the optical flow estimation matrix are *x*_(*u*, *v*)_ and *y*_(*u*, *v*)_, the pixel value corresponding to (*u*_0_, *v*_0_) should be:(1)$$f(u_{0},v_{0})=Q(u_{1},v_{1})*w_{1}+Q(u_{1},v_{2})*w_{2}+Q(u_{2},v_{1})*w_{3}+Q(u_{2},v_{2})*w_{4}$$
where *Q*(*u*, *v*) represents the value of pixel (*u*, *v*) in the key frame segmentation result and *w* represents the weight. The solution methods of *u*_1_, *u*_2_, *v*_1,_ and *v*_2_ are as follows:(2)$$\left\{\begin{array}{l} u_{1}=\left\lceil u_{0}-x_{\left(u_{0},v_{0}\right)}\right\rceil \\ u_{2}=\left\lfloor u_{0}-x_{\left(u_{0},v_{0}\right)}\right\rfloor \\ v_{1}=\left\lceil v_{0}-y_{\left(u_{0},v_{0}\right)}\right\rceil \\ v_{2}=\left\lfloor v_{0}-y_{\left(u_{0},v_{0}\right)}\right\rfloor \end{array}\right.$$

### 2.4. Optimizing Convolutional Layer 

After tracking the pixels by wrapping unit, ghost artifact is still a problem. Therefore, a convolutional layer was designed at the end of the optical flow estimation network. It was to generate an optimization matrix by using the high-level features of the optical flow estimation network, which can eliminate the object of time T and retain the object of time T + 1. The input of this convolutional layer is the output of last convolutional layer in the optical flow estimation network. The location of this convolutional layer is shown in Figure 7.

Finally, after calculation by Formula (3), the results can be obtained.
(3)$$prediction=M_{o}•M_{w}$$
where *M**_o_* is the output of optimizing convolution layer and *M**_w_* is the output of wrapping unit.

The loss function for optimizing the convolutional layer’s training is the same as semantic segmentation network’s. During training of optimizing the convolution layer, only parameters of optical flow estimation network and convolutional layer are updated, while parameters of semantic segmentation network are not updated. The DDFlow [16], an unsupervised optical flow estimation network training method, was adopted to carry out unsupervised training in this paper, and the training did not need any marked data sets. The specific training methods were as follows:

Step 1: Select two consecutive frames, denoted as F1 and F2.

Step 2: Input F1 into semantic segmentation network to get a robust result, denoted as R1.

Step 3: Input F2 and F1 into the optical flow estimation network for optical flow estimation. The optical flow estimation matrix is denoted as E. Combine E and R1 to get a tracking result, denoted as R2.

Step 4: Input F2 into the segmentation network to obtain a robust result of F2, denoted as R3.

Step 5: Take R2 as Prediction and R3 as Label to train the optimizing convolutional layer.

The training process is shown in Figure 8.

### 2.5. Adaptive Scheduling Network

As shown in Figure 9, a very slim adaptive scheduling network was designed for adaptive scheduling of the robust segmentation network and fast optical flow estimation module. The network consisted of a convolutional layer and three fully connected layers. The input of this network was the output of FlowNetS, which is a floating number. It can be set to any physically meaningful value and is used as a threshold for adaptive scheduling. If the output floating number is greater than the threshold value, the optical flow estimation module is used for lane segmentation. Otherwise, the robust segmentation network is used for lane segmentation. In this paper, the accuracy [10] was used as the threshold. The formula of accuracy is:(4)$$\mathrm{Acc}=\sum_{im}\frac{C_{im}}{S_{im}}$$
where *C_im_* is the number of correct pixels, a pixel is correct when its coordinate in the tracking result belongs to the coordinates of lane line in the original frame, and *S_im_* is the number of all pixels.

The loss function of the adaptive scheduling network adopts the most common mean square error. The training method of the adaptive scheduling network is as follows:

Step 1: Select two consecutive frames, denoted as F1 and F2.

Step 2: Input F1 into semantic segmentation network to get a robust result, denoted as R1.

Step 3: Input F2 and F1 into the optical flow estimation network for optical flow estimation. The optical flow estimation matrix is denoted as E. Combine E and R1 to get a tracking result, denoted as R2.

Step 4: Input F2 into the segmentation network to obtain a result, denoted as R3.

Step 5: Combine R2 and R3 to calculate the accuracy according to Formula (4), denoted as L.

Step 6: Record the FlowNetS output, which is in Step 3, denoted as D.

Step 7: Train the adaptive scheduling network with D as the network input and L as the label.

The training process of adaptive scheduling network is shown in Figure 10.

### 2.6. Mapping

In general, autonomous driving utilizes the polynomial in the camera coordinate system instead of the pixel coordinate system. Therefore, we proposed a method to map pixels from the pixel coordinate system to the camera coordinate system, based on [17,18].

Figure 11 is the corresponding diagram of the real world and the image plane. O is the camera position, E′ is the road vanishing point of the image plane, and E is the road vanishing point in the real world. S is the position of the object in the real world and S′ is the position of the object in the image plane. G′ is the optical center of the image plane and G is the position of the optical center of the image plane in the real world. C′ is a point on the road in the image plane and C is the position of C′ in the real world. The *d*_1_ is the longitudinal distance between C and camera O and *d*_2_ is the longitudinal distance between S and camera O.

Formula (5) can be deduced from Figure 11:(5)$$d_{1}=\frac{d_{2}\tan\alpha }{\tan\beta }$$

In Figure 11b, OG′ = *f*, where *f* is the camera focal length and the ordinates of G′, C′, S′ and E′ are denoted as *n*_0_, *n*_1_, *n*_2_, *n*_3_. Set the camera pixel focal length $\alpha_{y}=f/dy$ and it can be deduced from Figure 11:(6)$$\tan∠COG=\frac{C^{\prime }G^{\prime }}{OG^{\prime }}=\frac{n_{1}-n_{0}}{\alpha_{y}}$$
(7)$$\tan∠SOG=\frac{S^{\prime }G^{\prime }}{OG^{\prime }}=\frac{n_{0}-n_{2}}{\alpha_{y}}$$

Since E′ corresponds to the end of the road, OE is parallel to the road, and E′ corresponds to infinity in the camera coordinate system. Figure 11 can be derived:(8)$$\tan∠EOG=\frac{E^{\prime }G^{\prime }}{OG^{\prime }}=\frac{n_{0}-n_{3}}{\alpha_{y}}$$
(9)α=∠EOG−∠SOG
(10)β=∠EOG+∠COG

By substituting Formulas (6)–(10) into Formula (5), we can get: (11)$$d_{2}=d_{1}\frac{\left(n_{1}-n_{3})[\alpha_{y}^{2}+(n_{0}-n_{3})(n_{0}-n_{2})\right]}{\left(n_{2}-n_{3})[\alpha_{y}^{2}-(n_{0}-n_{3})(n_{1}-n_{0})\right]}$$

According to the above equations, as long as *d*_1_, $\alpha_{y},$ and *n*_2_ are calibrated, the vertical coordinates of pixels in the pixel coordinate system can be mapped to the camera coordinate system.

According to the above algorithm, we proposed an algorithm to map the abscissa of pixels from the pixel coordinate system to the camera coordinate system. In Figure 12, the ordinates of S′ and Q′ in the pixel coordinate system are *u*_2_ and *u*_4_, respectively.

According to the geometric relationship in Figure 12, we can easily deduce that:(12)$$\frac{OS^{\prime }}{S^{\prime }Q^{\prime }}=\frac{OS}{SQ}$$
(13)$$S^{\prime }Q^{\prime }=u_{4}-u_{2}$$
(14)$$OS^{\prime }=\sqrt{OG^{\prime 2}+G^{\prime }S^{\prime 2}}=\sqrt{\alpha_{y}^{2}+{\left(n_{0}-n_{2}\right)}^{2}}.$$

According to Figure 12, we can get that:(15)$$OS=\frac{d_{1}\tan\beta }{\sin\alpha }.$$

By combining Equations (16)–(19), it can be obtained:(16)$$\begin{array}{l} d_{3}=SQ=\frac{S^{\prime }Q^{\prime }}{OS^{\prime }}\times OS\\ =\frac{u_{4}-u_{2}}{\sqrt{\alpha_{y}^{2}+{\left(n_{0}-n_{2}\right)}^{2}}}\times \frac{d_{1}\tan(\arctan\frac{n_{0}-n_{3}}{\alpha_{y}}+\arctan\frac{n_{1}-n_{0}}{\alpha_{y}})}{\sin(\arctan\frac{n_{0}-n_{3}}{\alpha_{y}}-\arctan\frac{n_{0}-n_{2}}{\alpha_{y}})} \end{array}$$

If the object is on the left side of the camera, *d*_3_ is negative, and if not, it is positive.

## 3. Experiments 

In this section, three experiments were carried out. In Section 3.1, there is a comparison experiment about the lane segmentation model whose evaluation index includes processing time, accuracy, recall, and precision. In Section 3.2, the experiment mainly concentrated on the precision of lane discrimination. In Section 3.3, we carried out an experiment to verify the mapping algorithm. All experiments were carried out on a PC whose configurations included Inter(R)Core(TM)i7-7820X@3.60GHz, NVIDIA TITAN Xp, Python 3.7.0, and TensorFlow 1.12.0.

### 3.1. Experiment of Lane Segmentation Model

In order to verify the validity of the proposed lane detection model, the Tusimple data set and the self-collected data set were applied in the experiment. The Tusimple data set had high image clarity, no blur, and relatively simple detection difficulty. The self-collected data set came from a cheap and normal webcam whose images were with obvious occlusion, blurs, and poor illumination. The specific comparison is shown in Figure 13. The first column is the result of our method and the second column is the result of ENet. The first row is the result of Tusimple data set and the others are the results of the self-collected data set.

Figure 14 shows three representative working scenes. Figure 14a is the scene of an expressway ramp. At this time, vehicles drove with a low speed. The illumination was suitable with no obvious occlusions. When the threshold was set to 100, every frame was detected by semantic segmentation network. Without the acceleration of the optical flow tracking module, the accuracy was pretty high, but the speed was slow. When the threshold dropped continuously, the scheduled frequency of the optical flow tacking module increased continuously. It can be seen from the Figure 14a that the model proposed in this paper could increase the speed by three times while ensuring the accuracy. Figure 14b is an urban road scene. In order to test the robustness of the model, the vehicle changed lanes fast and continuously. It can be seen from the Figure 14b that the proposed model still increased the speed by two times while ensuring the accuracy. Figure 14c shows a narrow urban road, which is a more complex scene. In addition to an obvious lane on the left side of the vehicle, the middle lane was mostly blocked by the white car in front, and there was no obvious lane on the right side. There was also a row of vehicles with obvious edge features on the right side, which can easily cause incorrect detection. However, our model still enhanced the speed and ensurd the robustness.

We chose four indices to verify the performance of the lane detection model, which were accuracy, precision, and recall. The equation of accuracy is Formula (4). Precision’s [10] and recall’s [10] equations are as follows:(17)$$\mathrm{Precision}=\frac{TP}{TP+FP}$$
(18)$$\mathrm{Recall}=\frac{TP}{TP+FN}$$

True positive (*TP*) represents the number of correct lane pixels. False positive (*FP*) represents the number of wrong lane pixels. False negative (*FN)* represents the sum of wrong background lane pixels. Mean intersection over union (*MIoU*) can be used to evaluate the similarity between the original lane line and the segmentation lane line. It refers to TP, FP, and FN, which are defined as follows:(19)$$MIoU=\frac{1}{k+1}\sum_{i=0}^{k}\frac{TP}{FN+FP+TP}$$
where *k* is the number of pixels. If *MIoU* is 1, it means the lane line after segmenting is the same as the original one.

In order to experimentally compare with Deeplabv3plus [14] in the model in this paper, comparative analysis was also conducted with ENet [11], Bilateral segmentation network(Bisenet) [19], image cascade network (ICNet) [20], Deeplab [21], and pyramid scene parsing network (PSP Net) [22] on Tusimple and self-collected data set. The comparison results are shown in Table 1 and Table 2. It can be seen that our proposed method can balance accuracy and processing time. With keeping accuracy, our method is about three times faster than Deeplabv3plus, which can reach real-time detection. Compared with Tusimple data set, our self-collected data set was more complex. We collected the data with different weather, illumination, occlusion, and driving habits. As shown in Table 2, our proposed method can still keep a high accuracy comparing with ENet, Bisenet, ICNet, Deeplab, and PSPNet, which means our method is much more robust.

### 3.2. Experiment of Lane Discrimination

According to the lane segmentation results in Section 3.1, the algorithm in Section 2.4 was utilized to discriminate lanes. In order to verify the accuracy and robustness of the method, we made a comparison with Aslarry, Dpantoja, and LaneNet [10], which is one of the classic lane discrimination models based on CNN at present. The Tusimple data set and self-collected data including 5776 images were applied to the experiment. The self-collected data were sampled at noon, night, slopes, tunnels, curves, and on rainy days with about 152 kilometers running of the car. In addition, the colors of the lane line included white and yellow. The comparison results are shown in Table 3 and Figure 15. Indices are specifically defined as: accuracy = the number of correct detected frames/total detected frames. A frame is correct when the proportion of correct pixels in the discrimination result is larger than a certain value. Otherwise, it is a false frame. False detection rate (FDR) = sum of missed and false frames/total detected frames, for example, in 100 frames, two frames are false and three frames are not counted, then the FDR is 0.05. It can be seen that the two methods had similar accuracy on Tusimple data set, since this data set was collected with an easy scene. However, our proposed method was better working on our collected data set, which was collected with different weather, illumination, and occlusion. It means our proposed method is much more robust on lane discrimination.

### 3.3. Experiment of Mapping

In order to verify validity of our mapping method, we designed a distance estimation experiment. The evaluation index is described as follow:(20)$$Error=\left|\frac{TD-ED}{TD}\right|*100\%$$
where *TD* is the true distance, which is measured by a laser ranger finder and *ED* is the estimated distance. 

The internal and external parameters of the camera needed in the experiment were obtained by MATLAB calibration toolbox and confirmed that $\alpha_{y}$ = 780, *n*_1_ = 360, *d*_1_ = 7.00 m, *n*_0_ = 180, *n*_3_ = 173, and *u*_2_ = 320. The specific experimental steps were as follows:Lane segmentation;Lane discrimination;Mapping pixels from the pixel coordinate system to the camera coordinate system;Curving fitting based on least square method; andCalculating the distance and error;

The experiment was carried out in Chongqing University of Posts and Telecommunications. We measured the distance between two lane lines and confirmed the distance was about 2.5 m. Two lane curves, which were near to our vehicle, were selected to calculate their longitudinal distance and Error. The experimental results are shown in Table 4.

The fitting results are shown in the Figure 16. It can be found from the Figure 16 that there were large errors in the longitudinal distance from 0 m to 5 m, whether it was straightway or curved roads. The main reason is that the fitting highly relied on the position characteristics of pixel points mapping to the real-world coordinate system and the distance between the bottom of the image and the camera was 5 m, as shown in Figure 17, which means that the fitting result completely depended on the theoretical fitting result when the longitudinal distance was less than 5 m. Therefore, it will get a large error when the longitudinal distance is less than 5 m.

In order to theoretically analyze the relationship between the mapping error and the actual distance, we found all pixels that corresponded to longitudinal distance from 0 m to 80 m, as shown in Figure 18. Analyzing Figure 18, we can draw two conclusions: With the increase of the longitudinal distance, pixels get sparse. In other words, the longitudinal mapping error increases with the increase of longitudinal distance. The horizontal mapping error also increased with the increase of longitudinal distance. The points in Figure 18 are discrete because the value of the pixel is an integer.

## 4. Conclusions and Discussions

Lane segmentation model based on deep learning plays an important role in the field of autonomous driving and there are many related algorithms as well, some of which have high segmentation precision and some have fast segmentation speed. However, an algorithm with both accuracy and rapidity still remains a concern.

In this paper, we proposed a lane detection method based on semantic segmentation and optical flow estimation. In addition, in order to verify the performance of our method, we collected about 6000 images as a self-collected data set, which included various real-road conditions. The main innovation is summarized as: First, we sped up the robust semantic segmentation network by combining it with an optical flow estimation network. The experimental results verified that the whole model is three times faster than the robust semantic segmentation network at most, and the accuracy reduced by about 2% at most. Secondly, we proposed a mapping method to get lane curves in the camera coordinate system, which can provide feedback for autonomous driving. The experimental results indicated that our method can provide an effective feedback whose relative error is about 3.0%.

Although our model got a good robustness in lane line detection results, it still has some work we need to do in the future. Since the least square method was used for both straight and curve fitting, it will cause a certain error in the case of missing data, especially for curve fitting. Thus, how to reduce the fitting error is our next step for work.

## Figures and Tables

**Figure 1 sensors-21-00400-f001:**
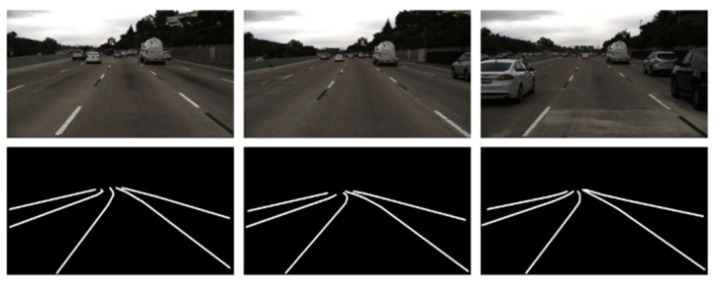
Continuous frames.

**Figure 2 sensors-21-00400-f002:**
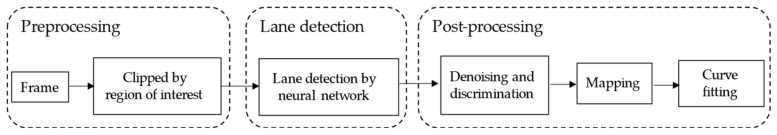
Lane detection block diagram.

**Figure 3 sensors-21-00400-f003:**
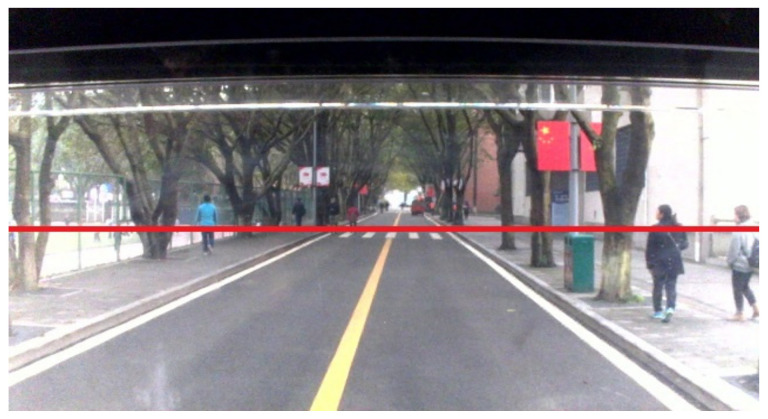
Region of interest.

**Figure 4 sensors-21-00400-f004:**
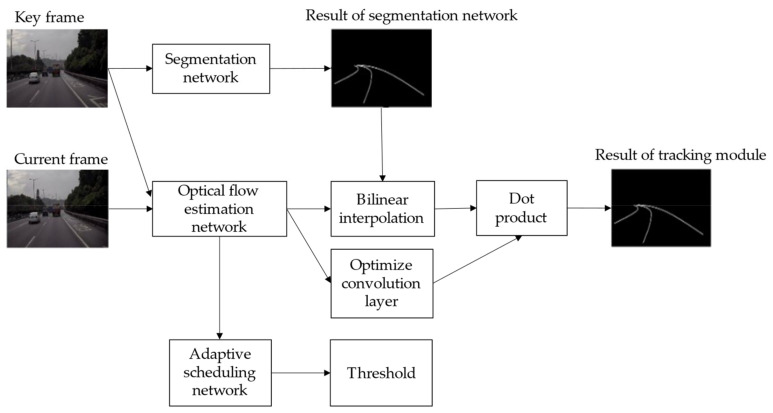
Pipeline of lane segmentation.

**Figure 5 sensors-21-00400-f005:**
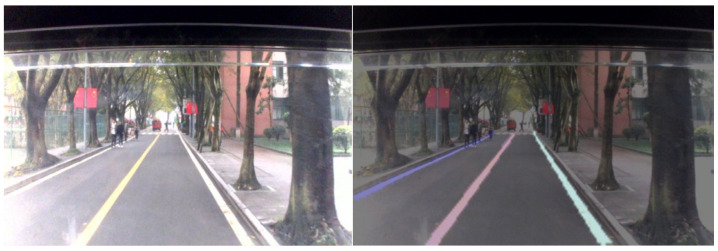
Effect of DBSCAN.

**Figure 6 sensors-21-00400-f006:**
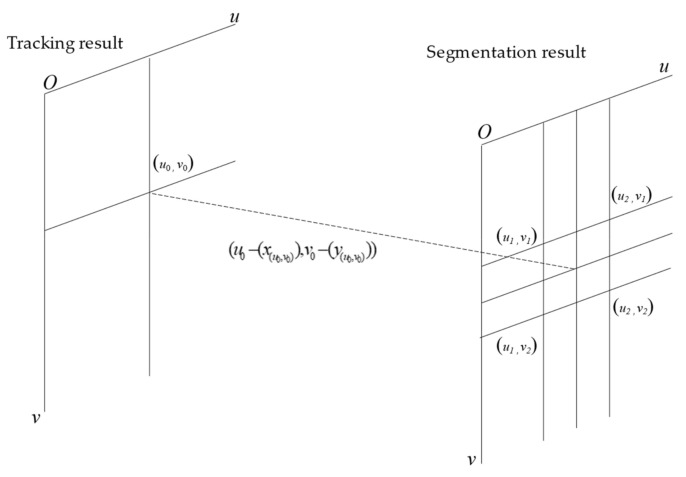
Operation schematic diagram of wrapping unit.

**Figure 7 sensors-21-00400-f007:**
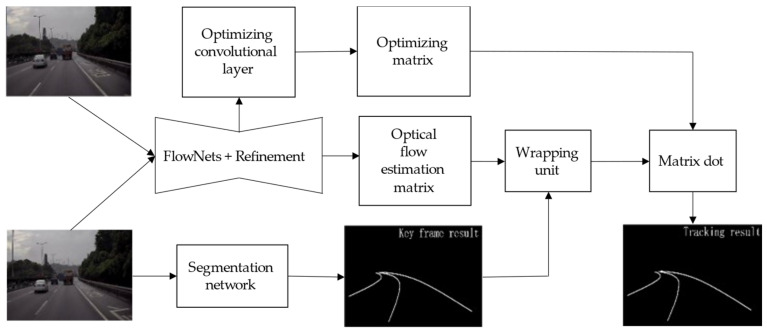
Location of optimizing convolutional layer.

**Figure 8 sensors-21-00400-f008:**
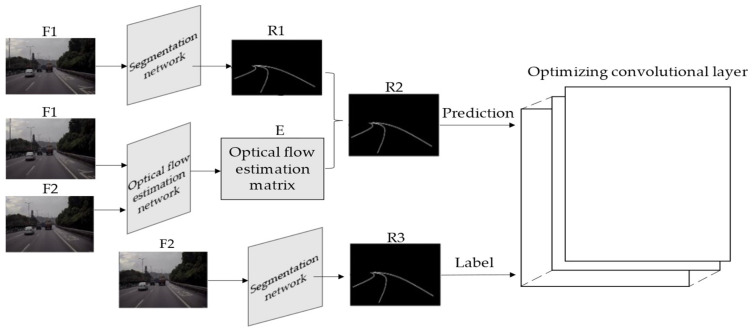
Training process of optimizing convolutional layer.

**Figure 9 sensors-21-00400-f009:**
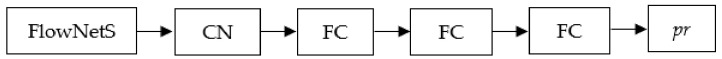
Adaptive scheduling network.

**Figure 10 sensors-21-00400-f010:**
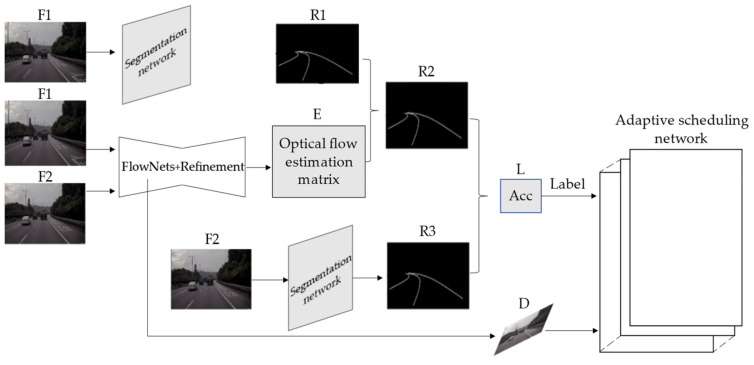
Training process of adaptive scheduling network.

**Figure 11 sensors-21-00400-f011:**
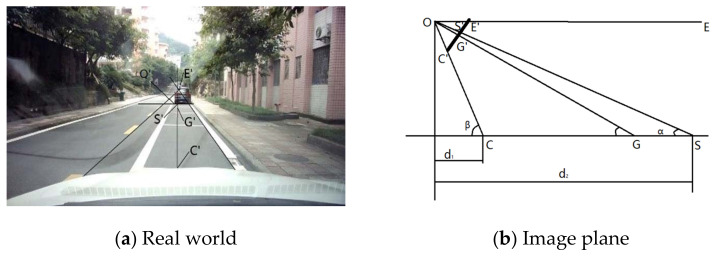
The real world and the image plane.

**Figure 12 sensors-21-00400-f012:**
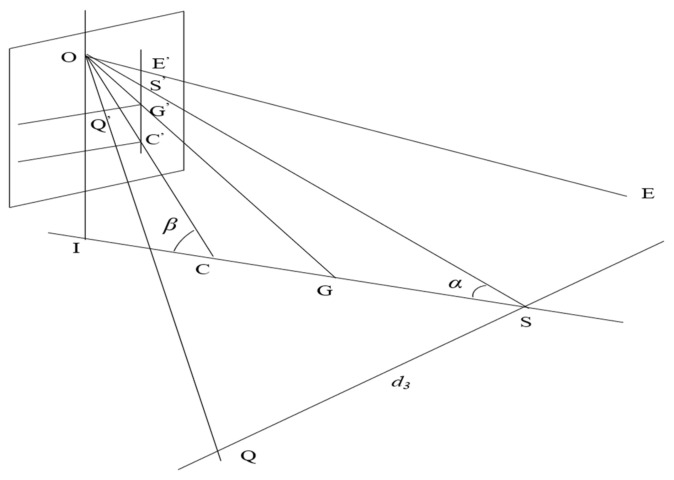
Abscissa mapping diagram.

**Figure 13 sensors-21-00400-f013:**
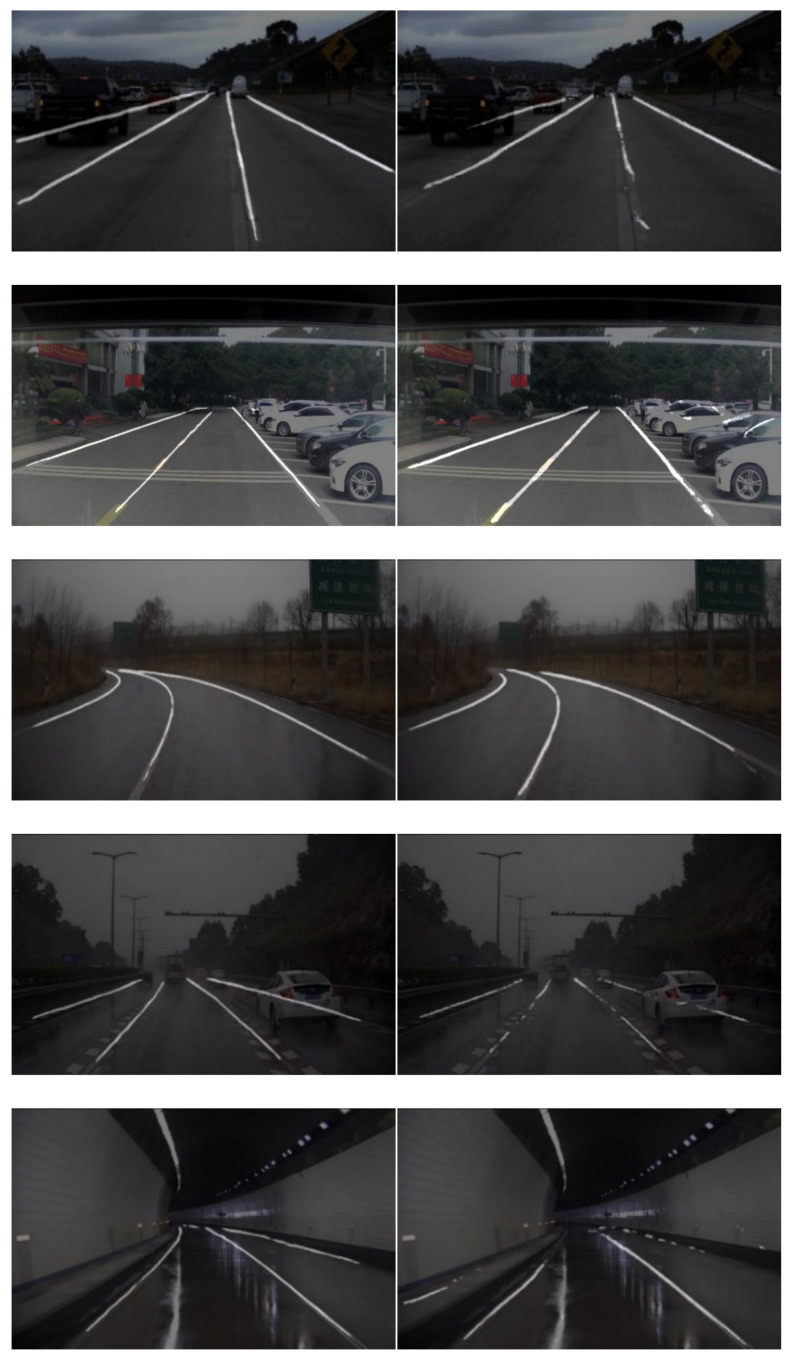
Comparison between our method and ENet.

**Figure 14 sensors-21-00400-f014:**
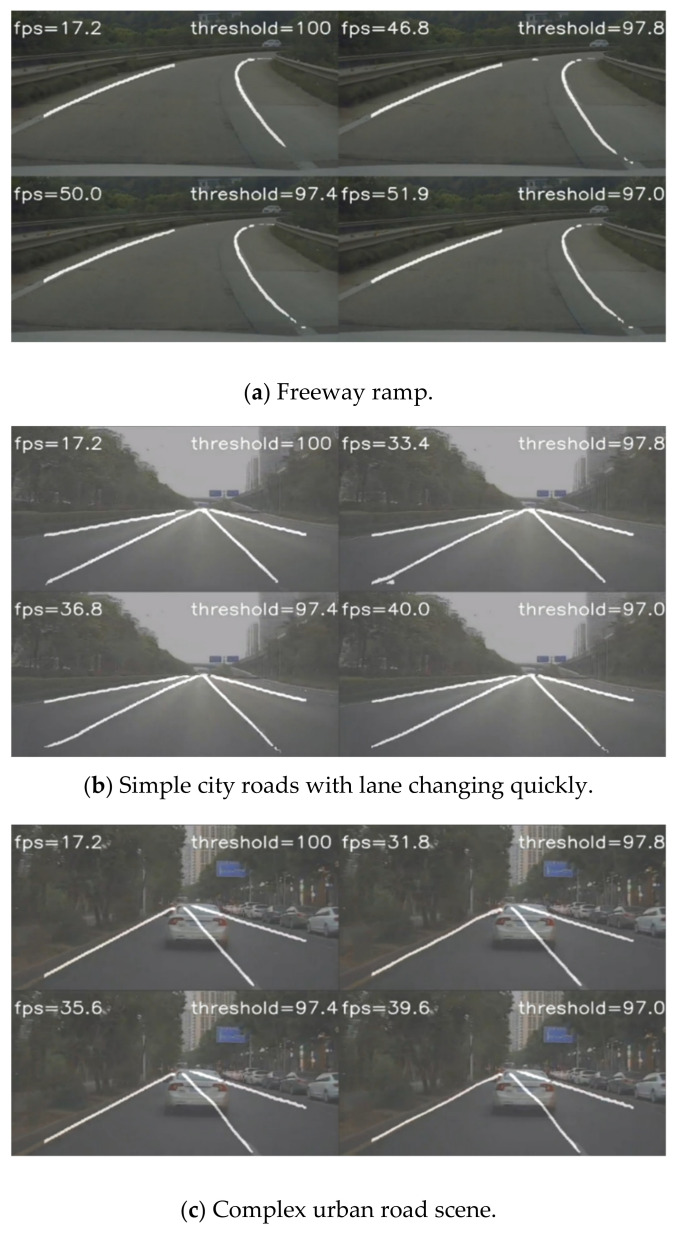
Compared with segmentation network.

**Figure 15 sensors-21-00400-f015:**
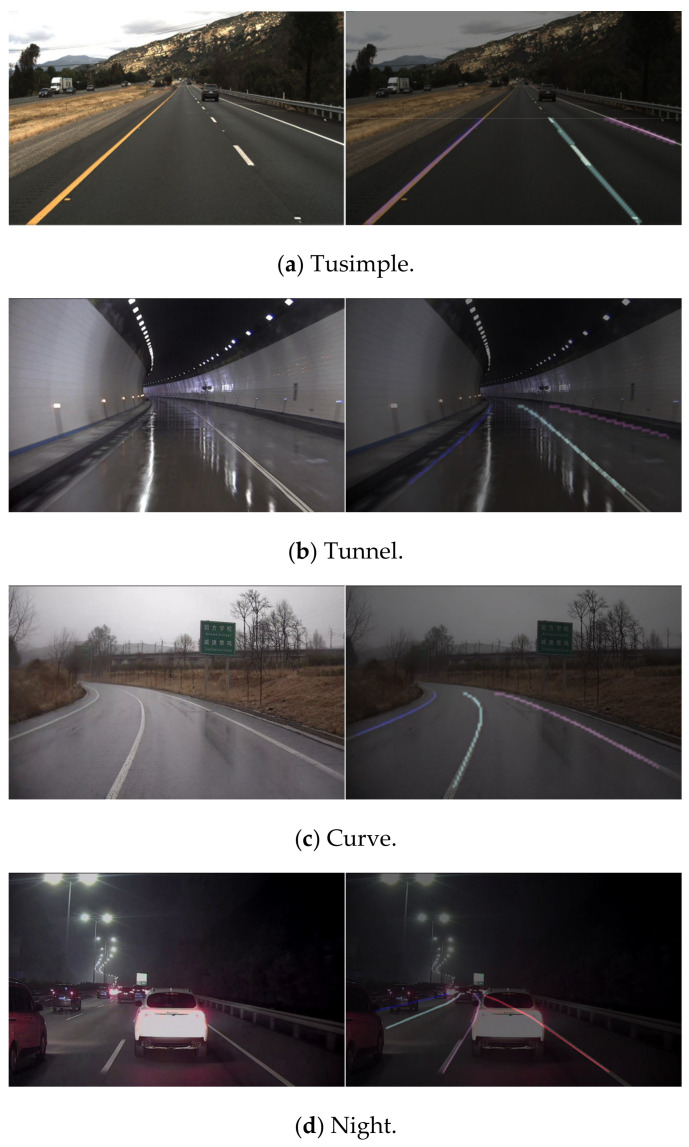
Effect of lane discrimination.

**Figure 16 sensors-21-00400-f016:**
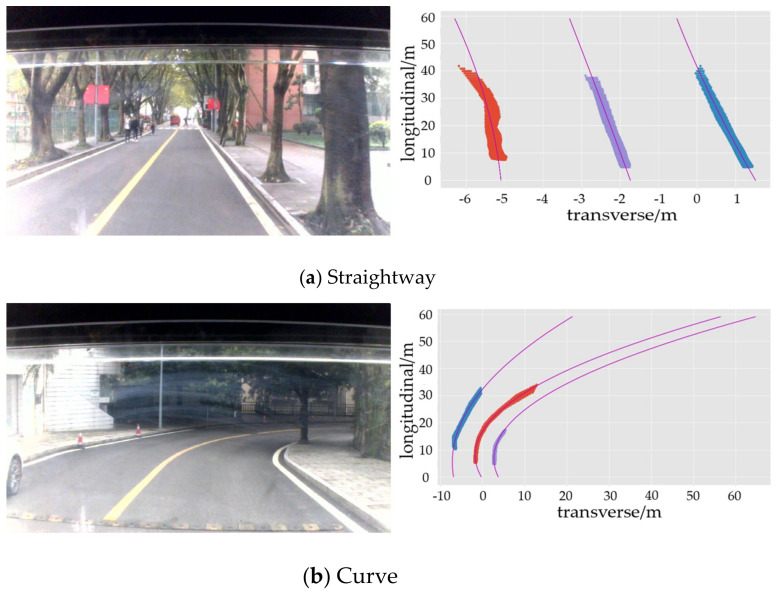
Effect of mapping.

**Figure 17 sensors-21-00400-f017:**
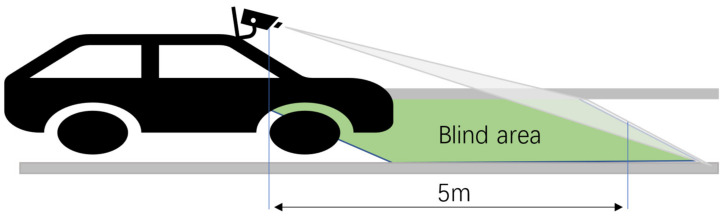
The blind area of the camera.

**Figure 18 sensors-21-00400-f018:**
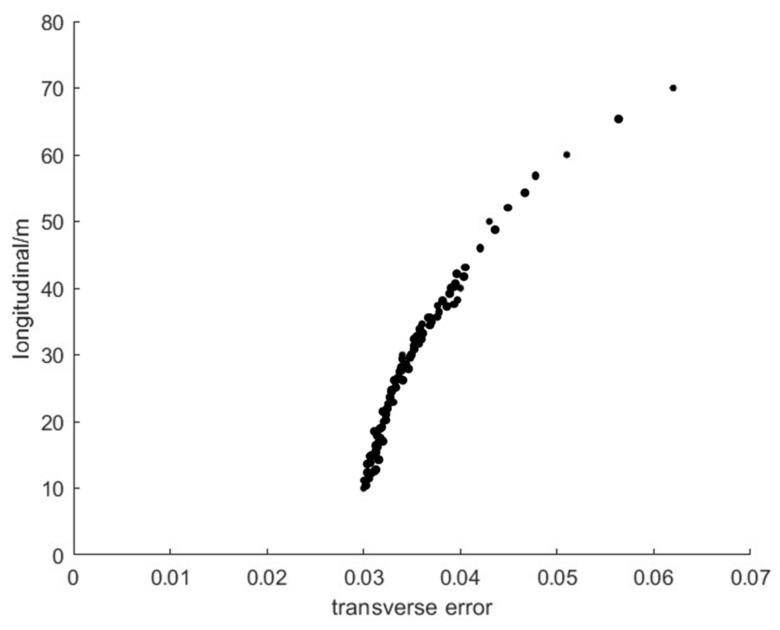
Relationship between longitudinal distance and transverse error.

**Table 1 sensors-21-00400-t001:** Result of Tusimple data set.

Method	Acc%	Precision/%	Recall/%	MIoU/%	Fps
ENet [11]	96.2	88.2	95.2	81.1	135
Bisenet [19]	96.1	88.5	95.8	81.3	106
ICNet [20]	96.3	88.4	96.1	81.8	39
Deeplab [21]	96.2	88.9	96.0	82.6	30
PSP Net [22]	96.4	90.2	96.4	85.6	21
Deeplabv3plus [14]	96.5	90.6	96.5	87.7	17
Proposed method-threshold (0.978)	96.4	90.0	96.1	85.9	47
Proposed method-threshold (0.970)	96.3	89.5	95.6	85.2	52

**Table 2 sensors-21-00400-t002:** Result of self-collected data set.

Method	Acc/%	Precision/%	Recall/%	MIoU/%	Fps
ENet [11]	87.8	75.4	75.9	72.3	135
Bisenet [19]	88.1	76.4	75.4	73.1	107
ICNet [20]	88.5	77.1	77.8	74.0	39
Deeplab [21]	88.8	78.8	78.4	74.9	30
PSP Net [22]	90.2	84.5	85.2	76.1	21
Deeplabv3plus [14]	94.5	88.3	87.6	80.1	17
Proposed method-threshold (0.978)	93.9	86.6	85.4	76.5	47
Proposed method-threshold (0.970)	93.1	85.9	85.1	76.2	52

**Table 3 sensors-21-00400-t003:** Result of lane discrimination.

Dataset	Method	Accuracy/%	FDR/%
Tusimple	Proposed method	94.2	5.8
	Aslarry	95.6	4.4
	Dpantoja	95.2	4.8
	LaneNet	95.5	4.5
Self-collected	Proposed method	90.0	10.0
	Aslarry	84.1	15.9
	Dpantoja	83.8	16.2
	LaneNet	84.3	15.7

**Table 4 sensors-21-00400-t004:** Result of mapping.

Longitudinal Distance/m	Transverse Error/%
0	7.6
10	3.0
20	3.2
30	3.4
40	4.0
50	4.3

## Data Availability

Publicly available datasets were analyzed in this study. This data can be found here: [https://github.com/TuSimple/tusimple-benchmark/issues/3].

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
