# Peer review of "A Fast and Robust Lane Detection Method Based on Semantic Segmentation and Optical Flow Estimation"

_sensors, 2021, doi:10.3390/s21020400_

Round 1
Reviewer 1 Report
The authors present an interesting paper about lane detaction based on semantic segmentation and opticla flow. They present a new method/approach whose results are very promising. However, it would be advisable to proceed with the following improvements:
1.- English writing review, as there are some typos.
2.- regarding to the comparative analysis between the method proposed by the authors and the other methods, is only performed in relation to which it present better results..
3.- It would be interesting to know the opinion of the authors about the situations/reasons in which the proposed method does not work as well as the Deeplabv3plus method.
Finally, congratulations on your work.
Reviewer 2 Report
The paper deals with a quite well-studied and it provides another approach to improve performance. There are some topics that must be improved:
- State of the art review must be enhanced because there are several papers dealing withntopic thata re missing
- Contributions and more comparisons with previous studies must be included
- Experimental systems use this kind of information and the results are ok. Please, highlight the contributions.
- Results section is quite short and more results that show performance must be included. Section 3.1 is very descriptive and Section 3.2 does not provide enough results (in how many km has the system been tested, in which conditions, etc)
- English must be carefully revised
Reviewer 3 Report
The paper describes an approach to road lane detection based on semantic segmentation and optical flow estimation. The main aim of the research is to provide an algorithm that can improve the detection speed while preserving the robustness of the outcome.
The proposed framework consists of three key elements: lane segmentation, lane discrimination and mapping. Semantic segmentation and optical flow estimation are used in the lane segmentation stage, where an adaptive scheduling network is responsible for choosing which of the two method to apply. After the segmentation phase, the DBSCAN (Density-Based Spatial Clustering of Applications with Noise) algorithm is adopted to distinguish lane lines based on the pixel clustering results. Lastly, in the mapping phase, the pixels pertaining to the lane lines are transformed from a pixel coordinate system into a camera coordinate system.
I found the work interesting. In particular, the idea of increasing the speed of the segmentation task by treating various image frames by applying a different detection method depending on the 'complexity' of the image, represents an innovative element. Nevertheless, there are some points in the manuscript that need clarification:
- Although the text speaks of "lane detection", it seems that the proposed method focuses on detecting pixels belonging to the lines of the road lane, not the lane itself. It would therefore be more correct to speak of "lane line detection". Please clarify.
- Lines 50-51: the Authors speak of 'difficult' and 'simple' frames without explaining what they mean by this. The adaptive scheduling threshold is introduced later, but it would be helpful for the reader to explain the concept, which is not obvious, from the beginning.
- Line 167: please check Eq. 5. It should be Q(u0-x(...),v0-y(...)) = ... Also, equations from 3 to 6 seem to be redundant. Are the Authors just describing how to perform a 2D interpolation? Or is there any added value in this for the paper?
- Lines 189-195: please add a diagram (possibly adapt Fig. 7) referring to the described steps and containing the elements (F, R, E) mentioned in each step. This will facilitate reading and understanding.
- Line 199: please correct full connection layers to fully connected layers.
- Line 205: when is a pixel considered correct?
- Lines 210-219: see point 4 (Lines 189-195).
- Line 221: please correct automatic to automated (or autonomous).
- Line 221: please correct utilize to utilizes.
- Line 292: looking at Fig. 12(b), no lane change is evident. Please provide a better image or clarify the example.
- Lines 308-309: please be more precise in the definitions of True Positive (TP), False Positive (FP) and False Negative (FN). Especially FN.
- Lines 331-332: Confusing definitions. Please define accuracy and False Detection Rate (FDR) in terms of TP, TN, FP, FN. Also, please provide a brief explanation of the meaning of FDR before providing the formula.
- Line 359: “distance between two lanes” should be “distance between two lane lines”.
- Lines 371-372: not clear; do the Authors mean that the horizontal mapping error increases as the horizontal distance increases? Please clarify and show data.
- Lines 373-375: please clarify Fig.15 and Tab. 4:
- regarding the chart in Fig. 15, the values on the horizontal axis appear to be errors as also stated in the caption. However, the axis is labelled as distance (transverse/m);
- at each given distance reported both in the chart in Fig. 15 and in Tab. 4, the error values according to the graph are larger than those shown in the table.
- Line 381: please correct accurate to accuracy.
- Line 382: see point 8 (Line 221).
- Line 384: according to the values shown in Tab. 4, the error should be about 3.5% (not considering the entry 7.6). Please check.
- Please consider proofreading English language.
Round 2
Reviewer 2 Report
My previous concerns have been solved but there is still one item to be considered:
- Please, discuss the representativeness of results. I mean, are the frames, number con kilometers, types of scenarios represnetantive enough? Are there any type of situation that this research cannot work propertly?
Reviewer 3 Report
The Authors elaborated on most of the points raised during the first review, enriching the overall quality of the paper. However, there are some aspects that still need clarification:
- Line 30: the added phrase is not an explanation.
- Lines 165-168: as stated by the Authors, equations 3 to 6 have the sole purpose of showing how to perform a bilinear interpolation. This confirms what was highlighted in the previous review, namely that they are redundant and do not add any value to the paper. I would recommend removing them.
- Line 185: please check and clarify the following: “The training need not to marked datasets.”
- Lines 283-286: according to answer n. 10 provided by the Authors, the lane change experiment was conducted to assess the robustness of the model in case of quick changes in speed; I have some concerns: at what speed was the experiment conducted? What is the speed variation during a lane change? I believe the test is not appropriate for the specified purpose.
- Lines 365-366: please provide a picture showing the positioning of the camera on the vehicle during the experiments and the 5 m distance mentioned in the text; this would also be useful to support the following statement: “the distance between the bottom of the image and the camera is 5m”.
- Line 367: please check and clarify the following: “Otherwise, it will get a large error”.
- Please consider proofreading by a native English speaker.
